# PDIA3 Expression Is Altered in the Limbic Brain Regions of Triple-Transgenic Mouse Model of Alzheimer’s Disease

**DOI:** 10.3390/ijms24033005

**Published:** 2023-02-03

**Authors:** Tommaso Cassano, Flavia Giamogante, Silvio Calcagnini, Adele Romano, Angelo Michele Lavecchia, Francesca Inglese, Giuliano Paglia, Vidyasagar Naik Bukke, Antonino Davide Romano, Marzia Friuli, Fabio Altieri, Silvana Gaetani

**Affiliations:** 1Department of Medical and Surgical Sciences, University of Foggia, Via L. Pinto 1, 71122 Foggia, Italy; 2Department of Biochemical Sciences “A. Rossi Fanelli”, Sapienza University of Rome, P.le Aldo Moro 5, 00185 Rome, Italy; 3Department of Physiology and Pharmacology “V. Erspamer”, Sapienza University of Rome, P.le Aldo Moro 5, 00185 Rome, Italy

**Keywords:** Alzheimer’s disease, disulfide isomerase isoform A3, limbic brain regions, chaperone, 3×Tg-AD mice, β-amyloid, ageing, longitudinal study, neuroinflammation

## Abstract

In the present study, we used a mouse model of Alzheimer’s disease (AD) (3×Tg-AD mice) to longitudinally analyse the expression level of PDIA3, a protein disulfide isomerase and endoplasmic reticulum (ER) chaperone, in selected brain limbic areas strongly affected by AD-pathology (amygdala, entorhinal cortex, dorsal and ventral hippocampus). Our results suggest that, while in Non-Tg mice PDIA3 levels gradually reduce with aging in all brain regions analyzed, 3×Tg-AD mice showed an age-dependent increase in PDIA3 levels in the amygdala, entorhinal cortex, and ventral hippocampus. A significant reduction of PDIA3 was observed in 3×Tg-AD mice already at 6 months of age, as compared to age-matched Non-Tg mice. A comparative immunohistochemistry analysis performed on 3×Tg-AD mice at 6 (mild AD-like pathology) and 18 (severe AD-like pathology) months of age showed a direct correlation between the cellular level of Aβ and PDIA3 proteins in all the brain regions analysed, even if with different magnitudes. Additionally, an immunohistochemistry analysis showed the presence of PDIA3 in all post-mitotic neurons and astrocytes. Overall, altered PDIA3 levels appear to be age- and/or pathology-dependent, corroborating the ER chaperone’s involvement in AD pathology, and supporting the PDIA3 protein as a potential novel therapeutic target for the treatment of AD.

## 1. Introduction

Alzheimer’s disease (AD) is a neurodegenerative disorder that is clinically associated with progressive cognitive impairment [1]. AD brain histology is characterized by the accumulation of amyloid β-peptide (Aβ) plaques and neurofibrillary tangles (NFTs) composed of hyperphosphorylated tau protein [1].

Currently available medications only offer limited and transient benefits to a small proportion of AD patients; therefore, there is an urgent need for the development of therapeutic strategies that target novel mechanisms.

The stress response of the endoplasmic reticulum (ER) is considered a crucial process in the etiopathology of AD (for review, see [2]). The increase of pathogenic aberrant proteins and the dysregulation of intracellular calcium homeostasis are key processes underlying the induction of ER stress, which leads to cell death. Several *in vitro* studies have suggested that Aβ oligomers or fibrils induce ER stress in primary cultures of neuronal cells, a variety of different cell lines, and in brain slices [3,4,5]. Moreover, further investigations aimed to understand the molecular mechanisms that underlie the connection between extracellular Aβ and intracellular ER. The most likely mediator between Aβ levels and ER stress is calcium; one of the proposed hypotheses is that Aβ binds glutamatergic receptors that in turn are able to induce ER stress-dependent cell death by altering cytosolic calcium homeostasis [6].

In this scenario, a pivotal role is played by the mechanisms promoting the clearance of neurotoxic and/or misfolded proteins, also representing an approach that may limit the onset and slow the progression of AD. Among these mechanisms, chaperone-mediated autophagy (CMA), which involves the translocation of non-membrane bound, chaperone-captured substrates across the lysosomal membrane, is an interesting target for potential therapies.

A variety of studies are exploring the involvement of the protein disulfide isomerase isoform A3 (PDIA3) in the response to several types of stress in different neurodegenerative diseases, including AD, Parkinson’s disease, and Prion Diseases [7,8,9,10,11].

PDIA3, also known as ERp57, GRP58, and 1,25D3-MARRS, is a member of the protein disulfide isomerase family, composed of 505 amino acids, with a molecular weight of 57 kDa, and a structure characterized by four thioredoxin-like domains: *a*, *b*, *b*’ and *a*’ [12,13,14]. PDIA3 is predominantly localized in the ER, where it is involved in the correct folding of newly synthetized glycoproteins and in the assembly of the major histocompatibility complex (MHC) class I molecules, maintenance of calcium homeostasis, endoplasmic-reticulum-associated protein degradation (ERAD), and modulation of proliferation and apoptosis through the unfolded protein response (UPR) [15,16,17,18,19,20]. PDIA3 is also present in the cytosol, where it can interact, among other proteins, with the mammalian target of rapamycin (mTOR), which results up-regulated in selected neurons of AD brains that are predicted to develop tau pathology [21]; finally, PDIA3 is also present in the nucleus, where it can directly bind to DNA regions rich in A/T, regulating gene expression [22]. Furthermore, many studies have showed that PDIA3 can be found on the cell surface, where it binds 1α, 25-dihydroxycholecalciferol, followed by the activation of a fast response pathway [23,24].

The available evidence indicates multiple distinct functional roles for PDIA3 under physiological and disease states. In particular, PDIA3 seems to play a role in cell protection against oxidative stress through its redox and chaperone activities. During cellular stress PDIA3, as well as other PDIs, can prevent the neurotoxicity associated with ER stress and protein misfolding, and the development of diseases related to unfolded/misfolded proteins’ accumulation, including Aβ [11,25,26,27]. On the other hand, it has also been reported that PDIA3 chemical modification, induced by NO or reactive oxygen species, can negatively influence its putative protective effect. In fact, it has been reported that S-nitrosylation of PDIs results in protein inhibition and leads to ER stress, which can induce apoptosis [28,29,30], while the oxidation of PDIA3 was reported to be associated with AD [31].

Although the role for PDIs in contributing to AD pathology has been supported by an increased expression of PDIs in AD brains and their co-localization with neurofibrillary tangles in AD patients [32], pathophysiologically relevant amounts of S-nitrosylated PDIs were also noted [33,34,35].

In line with the latter observation, it has been reported that diosgenin, a plant-derived steroidal saponin structurally similar to calcitriol, an endogenous PDIA3 ligand, acts as an exogenous activator of PDIA3, improving memory performance in the object recognition test by reducing amyloid plaques and neurofibrillary tangles in the cerebral cortex and hippocampus of 5XFAD mice, an engineered mouse model of AD harboring five familial AD mutations [36]. Likewise, the diosgenin derivative caprospinol (diosgenin 3-caproate), reduces amyloid deposits and improves memory dysfunction in Aβ_1-42_-infused rats, another preclinical AD model [37]. Furthermore, it has been demonstrated that PDIA3 expression levels significantly increased after Aβ_1-42_ treatment in HMO6 cells, an immortalized human microglial cell line, and in microglial cells from 5XFAD mouse brains [38]. Recently, PDIA3 has been reported to protect neuronal cells from Aβ-induced toxicity [39]. These observations strongly suggest an important link between PDIA3 signaling and AD; however, despite these promising observations, data are still sparse, and the relationship between alterations of PDIA3 expression and the development of AD neuropathology remains unclear.

Therefore, the aim of the present study was to evaluate whether brain PDIA3 expression is altered in a murine triple transgenic model of AD (3×Tg-AD) in comparison with their wild-type littermates (Non-Tg). Consequently, by studying the temporal expression of PDIA3 in Non-Tg mice, our study allowed us also to analyze the impact of aging on PDIA3 expression. 3×Tg-AD mice, which harbor three mutant human genes [40], mimic the critical aspects of AD-neuropathology observed in human AD patients [40,41,42,43,44,45]. To investigate whether the temporal and regional patterns of such possible alterations may overlap with those of Aβ and tau pathology in this AD model, brain PDIA3 expression was analyzed at two different stages (mild and severe) of AD-like pathology [40]. In particular, the brains of 6- (mild pathology) and 18- (severe pathology) month-old mice were analyzed by dot blot analysis and double-immunofluorescence, followed by the semi-quantitative analysis of the respective signals. We focused our investigations on the amygdala, hippocampus (dorsal and ventral), and entorhinal cortex, all brain regions strongly affected by the AD-pathology, and whose alterations have major functional impacts on AD symptoms. A triple-immunostaining was also performed to investigate the sub-cellular localization of PDIA3.

Taken together, the expression and the localization pattern of PDIA3 may help us to highlight the role of PDIA3 in both aging and Alzheimer’s condition, thus supporting the idea that it might represent a novel molecular target for the development of a more efficacious pharmacological approach to AD pathology.

## 2. Results

### 2.1. Alteration of PDIA3 Protein Expression in the Limbic Brain Regions

To evaluate the impact of aging and genotype on PDIA3 expression, different brain regions of the limbic system (amygdala, entorhinal cortex, dorsal and ventral hippocampus) were evaluated in both adult and aged Non-Tg and 3×Tg-AD mice by dot blot assay. An overall analysis by two-way ANOVA, with genotype (3×Tg-AD vs. Non-Tg) and age (6 months of age vs. 18 months of age) as between-subject factors, indicated that both factors may contribute to the alteration of PDIA3 protein expression in the limbic brain regions (Table 1).

To better define the aging and genotype contribution to PDIA3 expression, mice groups with the same genotype and differing by age, or mice groups differing both in genotype and age, were compared.

As first, the impact of aging on PDIA3 expression has been evaluated in adult and aged Non-Tg mice. We observed a significant decrease of PDIA3 levels in the amygdala (−38%, *p* < 0.05, Figure 1A), dorsal hippocampus (−20%, *p* < 0.001, Figure 1C) and ventral hippocampus (−40%, *p* < 0.01, Figure 1D) of 18-month-old Non-Tg mice, as compared to 6-month-old Non-Tg mice; a trend toward a decrease was also observed in the entorhinal cortex (−14%, n.s., Figure 1B).

To investigate whether the progression of AD-like pathology may be paralleled by an altered expression of PDIA3 over time, we evaluated its levels at the mild and severe stages of the disease. Interestingly, 18-month-old 3×Tg-AD mice showed a significant increase of PDIA3 levels in the amygdala (+130%, *p* < 0.01, Figure 1A) and entorhinal cortex (+38%, *p* < 0.05, Figure 1B), while only a trend toward an increase was observed in the ventral hippocampus (+31%, n.s., Figure 1D), as compared to 6-month-old 3×Tg-AD mice; surprisingly, a significant decrease was observed in the dorsal hippocampus (−34%, *p* < 0.001, Figure 1C).

Finally, to evaluate the effect of mild AD-pathology on the PDIA3 levels, 6-month-old 3×Tg-AD mice were compared to age-matched Non-Tg mice. A statistical analysis revealed that PDIA3 expression was significantly reduced in the amygdala (−55%, *p* < 0.01, Figure 1A), entorhinal cortex (−41%, *p* < 0.001, Figure 1B), and ventral hippocampus (−46%, *p* < 0.01, Figure 1D) of 6-month-old 3×Tg-AD mice; surprisingly no significant effect was observed in the dorsal hippocampus (Figure 1C).

### 2.2. Aβ/APP-PDIA3 Double-Fluorescent Immunostaining

Since PDIA3 is deregulated in many neurodegenerative diseases, we conducted a longitudinal study to evaluate whether the altered PDIA3 expression observed over time in 3×Tg-AD mice by a dot blot analysis, was related to a different cellular distribution in the limbic areas analyzed. Additionally, a comparison with the expression and distribution of Aβ/APP was carried out. Therefore, we performed a double-fluorescent immunostaining on the selected limbic areas of both 6- and 18-month-old 3×Tg-AD mice, referred to as the mild and severe pathology groups, respectively.

Representative microphotographs of Aβ/APP-PDIA3 double-fluorescent immunostaining (green and red stain, respectively) and a scatterplot analysis of Aβ/APP vs. PDIA3 protein levels in the amygdala, entorhinal cortex, and dorsal and ventral hippocampus brain regions are shown in Figure 2, Figure 3, Figure 4 and Figure 5, respectively. In Table 2 the results from the Pearson’s correlation analysis of the data obtained from double immunofluorescent staining are summarized.

As previously reported [43,44,46], we observed in 3×Tg-AD mice an age-dependent increase of the Aβ levels in all brain areas investigated in this study. Interestingly, a scatterplot analysis indicated a direct correlation between PDIA3 expression and the build-up of Aβ pathology, but with different magnitudes among the different considered brain regions. A Pearson correlation test revealed a strong positive correlation in the basolateral amygdala (r = 0.8132, *p* < 0.001; Figure 2C) and the dorsal CA1 region of the hippocampus (r = 0.8457, *p* < 0.001; Figure 4C), whilst a moderate but still positive correlation was observed in the entorhinal cortex (r = 0.6606, *p* < 0.001; Figure 3C) and in the ventral CA1 region of the hippocampus (r = 0.3378, *p* < 0.01; Figure 5C).

### 2.3. PDIA3-NeuN-GFAP Triple-Fluorescent Immunostaining

Previous studies have reported that PDIA3 transcript is abundantly expressed in all cerebral cell types [47], therefore we profiled the pattern of PDIA3 expression in both NeuN-positive differentiated neurons and GFAP-positive astrocytes of 6- and 18-month-old 3×Tg-AD mice, referred to as the mild and severe AD-pathology groups, respectively. In particular, the qualitative analysis performed under the microscope allow us to distinguish the localization of our markers (PDIA3-positive cells: red; NeuN-positive differentiated neurons: green; GFAP-positive astrocytes: blue) in different limbic areas from 3×Tg-AD mice. Observing the subcellular expression patterns of PDIA3, it seems that its nuclear localization was increased in aged 3×Tg-AD mice, thus suggesting a possible redistribution of PDIA chaperones during the progression of the neuropathology ( 6A–9A). Moreover, we observed that PDIA3 (red) co-localized with NeuN-positive differentiated neurons (green) in all brain regions of 3×Tg-AD mice at both time points considered (Figure 6A, Figure 7A, Figure 8A, Figure 9A). Likewise, PDIA3 (red) co-localized with GFAP-positive astrocytes (blue), whose expression increased in an age-dependent fashion in the 3×Tg-AD mice (Figure 6A,B, Figure 7A,B, Figure 8A,B, Figure 9A,B; yellow arrows).

Interestingly, we also observed PDIA3-positive staining in a number of GFAP- and NeuN-negative cells from all the limbic areas considered in this study (Figure 6A, Figure 7A, Figure 8A, Figure 9A; white arrows). Based on the morphologies and dimensions of PDIA3-positive cells, we hypothesized that these cells may refer to microglial cells and/or immature/suffering neurons. To this regard, it has been previously demonstrated that NeuN staining can be altered or lost in immature and/or suffering neurons [48]. Future studies are needed to confirm such a hypothesis.

## 3. Discussion

The results of the present study demonstrate that PDIA3 expression is modulated in an age- and pathology-dependent fashion in the limbic brain regions of Non-Tg and 3×Tg-AD mice; moreover, we analyzed for the first time the expression pattern of PDIA3 in a number of brain cell populations derived from 6 month- (the stage of mild pathology) and 18 month-old (the stage of severe pathology) 3×Tg-AD mice.

Given the known potential roles of PDIA3 in aging, as well as in neurodegenerative diseases, we longitudinally analyzed the changes of PDIA3 protein levels in some limbic brain regions of Non-Tg and 3×Tg-AD mice, which develop both Aβ and tau pathologies in an age-dependent manner. To our knowledge, only a few studies have evaluated the PDIA3 expression in AD [11,31,36,38,49,50], but none of them have performed a longitudinal study in an animal model of AD.

The first interesting result obtained from the present study is the reduction of the PDIA3 protein level in the amygdala, entorhinal cortex, and hippocampus of Non-Tg mice at 18 months of age as compared to 6-month-old mice. Moreover, except for in the dorsal hippocampus, 6-month-old 3×Tg-AD mice showed lower levels of PDIA3, as compared to age-matched wild-type littermates. Altogether these data suggest that per se PDIA3 levels gradually reduce with age, and such a reduction might be accelerated by the occurrence of an AD-like pathology. Indeed, an age-related failure of the complex systems responsible for handling protein misfolding, results in the accumulation of misfolded and aggregated proteins, and consequent conformational diseases [6]. In this context, our data are in line with previous studies showing the effects of aging on chaperones’ concentrations in the ER of rat hepatocytes [51]. In particular, the authors clearly indicate that, even in the liver (an organ whose functional capacity is well-conserved throughout life) there are significant and specific declines associated with aging, which are due to a specific loss of the capacity of the ER chaperones to fold nascent proteins into their functional configurations [51]. This assumption is consistent with the concept that a major factor in the physiological deficits seen with aging could be due to a decreased capacity of the ER chaperones to process newly synthesized membrane and secretory proteins.

The ER stress response is regarded as an important process also in the etiology of AD [2]. The accumulation of misfolded proteins is considered a fundamental mechanism that underlies the induction of ER stress, leading to neuronal cell death [6]. As described by several authors, Aβ peptides are also produced in physiological conditions during the post-translational processing of the amyloid precursor protein (APP) [52,53,54,55], but their levels are kept low by the protein quality control mechanisms [56,57,58,59]. During aging, but mainly in neurodegenerative disorders such as AD, these proteostasis mechanisms become less efficient [51,60]. Therefore, free Aβ monomers are more prone to self-aggregate into amyloid fibrils and thus to form insoluble deposits, which are the major constituent of the amyloid plaques. The ER chaperones are a group of proteins acting as carriers to keep the Aβ in solution, thus avoiding its deposition [2,5,28,61,62]. To this regard, Erickson and colleagues observed, in the cerebrospinal fluid of healthy individuals, that most of the Aβ peptides produced during the APP processing were sequestered by PDIA3 and calreticulin chaperones [11]. These findings suggest that PDIA3, by acting as a molecular carrier for Aβ monomers, may prevent their self-aggregation and, in turn, the formation of insoluble aggregates. Therefore, the authors suggest that Aβ deposition during AD may be due to a faulty post-translational processing of APP by ER chaperones, which fail to form complexes with Aβ monomers [11]. In support of this hypothesis, it has been demonstrated that pharmacological stimulation of cortical and hippocampal axonal regrowth in transgenic mice co-expressing multiple AD-related mutations, is dependent upon the activation of PDIA3, which results in an overall reduction of AD neuropathology and increased performance in object recognition tasks in these murine models [36]. In this regard, we could hypothesize that the significant reduction of PDIA3 expression observed in the limbic brain regions of 3×Tg-AD mice at 6 months of age, as compared to age-matched Non-Tg mice, may underlie and/or contribute to the formation of intracellular Aβ oligomers.

A possible explanation for the observed reduced level of PDIA3 in the limbic brain regions of 3×Tg-AD mice at 6 months of age, as compared to age-matched Non-Tg mice, is the role of PDIA3 as a molecular carrier for Aβ monomers, preventing the formation of insoluble aggregates. Aβ peptides produced during the APP processing, and sequestered by PDIA3 and calreticulin chaperones, can be eliminated through the cerebrospinal fluid [11], thus reducing PDIA3 levels. Furthermore, it has been shown that short-term Aβ25-35 treatment of human neuroblastoma cells induces PDIA3 decreases in intracellular protein levels, different intracellular localization, and PDIA3 secretion in the cultured medium [39].

Therefore, as previously reported [36], our data further support the view that exogenous activation of PDIA3 may be a promising therapeutic option in the early phase of AD.

As 3×Tg-AD mice develop both Aβ and tau pathologies in an age-dependent manner, we investigated the expression of PDIA3 over time, comparing the levels of PDIA3 at 6 months of age versus 18 months of age. The latter is characterized by an extensive Aβ plaque burden and tau pathology, along with signs of activated microglia and inflammation [40,41,63]. Interestingly, 18-month-old 3×Tg-AD mice showed a significant increase of PDIA3 levels in the amygdala and entorhinal cortex compared to 6-month-old 3×Tg-AD mice, whilst a trend toward an increase was observed in the ventral hippocampus. These results demonstrated that the significant increase of PDIA3 expression is paralleled by the progression of AD-like pathology in an age-dependent manner. Further confirmation of these findings was obtained by the scatterplot of Aβ protein levels versus PDIA3 protein levels showing a direct correlation in all the brain regions considered in our study (Figure 2C, Figure 3C, Figure 4C, Figure 5C and Table 2). The severe AD-like pathology in 18-month-old 3×Tg-AD mice is characterized by glutamatergic alterations and mitochondrial impairment, as well as by marked activation of microglia accompanied by elevated mTOR protein levels and activation [44,63,64].

The mTOR signaling pathway has received much attention for its role in neurodegenerative disorders. In particular, mTOR was reported to be up-regulated in selected neurons of AD brains that are predicted to develop tau pathology, suggesting that chronic high levels of mTOR signaling may exert detrimental effects in AD brains [41,65,66,67,68]. In accordance with these findings, several studies focused on AD research have demonstrated that Aβ can enhance mTOR signaling, while rapamycin and its analogs, which act as mTOR complex 1 (mTORC1) inhibitors, significantly reduce intracellular Aβ levels [41,69,70,71]. Qian and colleagues have demonstrated that mTORC1 serves as a sensor of protein misfolding, helping to maintain the right balance of protein synthesis and degradation [72].

PDIA3, a redox-sensitive molecular chaperone, is involved in the redox-sensing mechanism by which mTORC1 responds to changes in the cellular redox conditions. In particular, it has been demonstrated that PDIA3 facilitates the assembly of mTORC1 and its ability to sense oxidizing agents [73]. Therefore, the neuronal stress conditions, triggered by the severe pathology in 18-month-old 3×Tg-AD mice, could increase the PDIA3 expression levels as well as the PDIA3-dependent redox-sensor activity on mTORC1, further contributing to its dysregulation, and thus worsening the AD-like neuropathology.

The mechanism by which PDIA3 regulates the assembly of mTORC1 could be dependent on its ability to catalyze the formation and/or isomerization of disulfide bonds [74,75,76]. In this regard, Sarvassov and Sabatini demonstrated that thiol oxidants decrease the interaction between Raptor and mTOR, whereas a reducing reagent stabilizes this complex [77]. In addition, it has been demonstrated that an elevated oxidative stress modifies mTORC1 and prevents its binding to the FKBP12-rapamycin complex, ultimately leading to rapamycin resistance [78]. Collectively, these studies suggest the regulation of the mTOR pathway by a redox-sensitive mechanism that can be based on the interaction between mTOR and PDIA3.

In line with this hypothesis, Ramírez-Rangel and colleagues demonstrated, by in vitro analyses performed on HEK293T and COS-7 cells, that PDIA3 overexpression was able to interfere with the signaling pathway of mTOR [73]. In particular, they observed that PDIA3 overexpression was able to increase the levels of mTORC1 and its activity. Moreover, they showed that in the presence of oxidizing agents, PDIA3 interacted with mTORC1 by acting as a critical component of a redox-sensing mechanism, which was able to modulate the mTOR signaling pathway [73].

An association between several ER stress markers, as well as unfolded protein response (UPR) proteins and the accumulation of NFTs, has been observed in post-mortem brain tissues from tauopathy patients, with a positive relationship between the severity of the protein aggregation and the disease status [79]. Recently, it has been reported that a small molecule, SB1617, can suppress abnormal tau protein aggregation through PDIA3 inhibition and the enhancement of the protein kinase-like endoplasmic reticulum kinase (PERK) signaling pathway [80]. NFTs accumulation occurs during the late stage of AD pathology. Thus, the inhibition of PDIA3 may be an effective strategy for regulating tauopathies and modulating AD progression.

Finally, in our experimental paradigm, we profiled the pattern of PDIA3 expression in both astrocytes (GFAP-positive cells) and differentiated neurons (NeuN-positive cells) by performing a triple-fluorescent immunostaining in the same limbic areas. Interestingly, our results showed, for the first time, the immunostaining profile of PDIA3 in all post-mitotic neurons, in most of the GFAP-positive cells, but also in a number of GFAP- and NeuN-negative cells found in the considered limbic areas (Figure 6A,B, Figure 7A,B, Figure 8A,B, Figure 9A,B). In this regard, we hypothesized that those cells could be microglial cells and/or immature/suffering neurons [48]. These unrecognized cells appeared in both 6- and 18-month-old 3×Tg-AD mice (Figure 6A, Figure 7A, Figure 8A, Figure 9A, white arrows), which are respectively characterized by a mild and severe neuropathology. Therefore, these findings could be explained by the presence of neuroinflammatory processes associated with AD, or by the loss of NeuN staining, which can result in alterations to, or the loss of, immature and/or suffering neurons, as demonstrated by Lavezzi and colleagues [48]. Moreover, in support of the first hypothesis, it was also observed that Aβ secretion is correlated to the activation of microglial cells, which are generally recruited during chronic inflammatory processes, typical of AD [38,81]. In addition, it has also been observed that PDIA3 was over-expressed in microglial cells stimulated by Aβ, probably to help the folding of newly synthesized glycoproteins in the ER [38].

Because the expression profile of PDIA3 in the brains of 3×Tg-AD mice is still poorly understood, further investigations are required to prove our hypothesis about the involvement of PDIA3 in AD and its different modulation during the progression of the disease.

## 4. Materials and Methods

### 4.1. Animals

6- and 18-month-old male Non-Tg and 3×Tg-AD mice were used in this study. The 3×Tg-AD mice, harboring PS1_M146V_, APP_swe_, and tau_p301L_ transgenes, were genetically engineered by LaFerla and colleagues at the Department of Neurobiology and Behavior, University of California, Irvine. Colonies of 3×Tg-AD and Non-Tg mice were established at the vivarium of the Puglia and Basilicata Experimental Zooprophylactic Institute (Foggia, Italy). The 3×Tg-AD mice background strain is C57BL6/129SvJ hybrid, and genotypes were confirmed from tail biopsy. The housing conditions were controlled (temperature 22 °C, light from 07:00–19:00, humidity 50–60%), and fresh food and water were freely available. The study was performed in accordance with the guidelines released by the Italian Ministry of Health (D.L. 26/2014) and the European Directive 2010/63/EU. All efforts were made to minimize the number of animals used in the study and their suffering.

For the aim of the study, one cohort each of 6- and 18-month-old animals were sacrificed by cervical dislocation, the brains were rapidly excided and freshly dissected to isolate the entorhinal cortex, dorsal and ventral hippocampus, and amygdala following a previously published protocol [82]. Soon after collection, all tissue samples were frozen on dry ice and stored at −80 °C until dot blotting analysis was performed.

A second cohort of 6- and 18-month-old mice were intracardially perfused with saline, followed by fixative solution (4% paraformaldehyde in 0.1 M PBS, pH 7.4) at a flow rate of 36 mL min^−1^. Then their brains were collected, post-fixed in 4% paraformaldehyde solution for 48 h and immersed in a sucrose PBS solution (25% sucrose in 0.1 M PBS, pH 7.4) overnight at 4 °C. The brains were then snap frozen in 2-methylbutane (−50 °C) (59060; Sigma-Aldrich SRL, Milan, Italy) and stored at −80 °C until an immunohistochemistry study was performed.

### 4.2. Protein Isolation and Blotting Analysis

After collection of the cerebral areas, tissues were lysed in Ripa Buffer (50 mM Tris-HCl pH 7.4, NaCl 150 mM, 1% NP-40, 0.5% sodium deoxycholate, 2 mM EDTA, 0.1% SDS, 1 mM DTT, 2X Protease Inhibitor Cocktail, 2 mM sodium orthovanadate) and the extracted proteins were quantified by a Bradford assay. Directly, 5 µg of proteins were spotted on nitrocellulose membranes using a dot blot apparatus. The membranes were blocked with 1% *w*/*v* Bovine Serum Albumin (Sigma-Aldrich) in PBS. The membranes were incubated with anti-PDIA3 rabbit polyclonal primary antibody (ABE1032, Merk Millipore, Milan, Italy, 1:1000 dilution) for 60 min, washed with TBST (50 mM Tris-HCl, pH 7.5, 150 mM NaCl, Tween 20), and then incubated with anti-rabbit peroxidase-conjugated secondary antibody (Jackson ImmunoResearch, Cambridge, UK, 1:5000 dilution) for an additional 60 min. After washing in TBST, the membranes were developed by chemiluminescence with ECL substrates (Immunological Sciences, Roma, Italy) and the signal was detected by ChemiDoc™ Imaging Systems (BioRad, Segrate, Italy). After stripping in glycine solution, 0.1 M pH 3.0, for 15 min, and neutralization with PBS, the membranes were blocked with 1% *w*/*v* I-block (Invitrogen, Thermo Fisher Scientific, Monza, Italy) in TBS (50 mM Tris-HCl, pH 7.5, 150 mM NaCl) for preparing the membrane to the next detection. Then, the membranes were incubated with anti-β-actin mouse monoclonal primary antibody (A1978, Sigma-Aldrich, 1:2000 dilution) for 60 min, washed in 1% *w*/*v* I-block and incubated with anti-mouse alkaline phosphatase-conjugated secondary antibody (Jackson ImmunoResearch, 1:5000 dilution) for an additional 60 min. After washing in 1% *w*/*v* I-block in TBS solution, the membranes were stained by colorimetric detection using alkaline phosphatase substrates (Sigma-Aldrich). The PDIA3 and β-actin protein expression was analyzed using Image Lab™ Software (BioRad). The β-actin immunostaining was used as house-keeping protein for normalization.

### 4.3. Double-Immunofluorescence

For double fluorescence immunostaining, 20-µm-thick brain coronal sections were obtained using a cryostat (Microm™ HM550, Thermo Fisher Scientific, Ann Arbor, MI, USA), and were mounted on positively charged slides, which were stored at −20° C until being further processed. The brain sections were incubated with 90% formic acid for 7 min followed by PBS washes. Then, brain sections were blocked with a PBS solution containing 5% normal goat serum and 0.3% Triton X-100, followed by overnight incubation with purified anti-β-amyloid/APP 1–16 monoclonal primary antibody (6E10, 803002, BioLegend^®^, San Diego, CA, USA, 1:1500 dilution) and with anti-PDIA3 rabbit polyclonal primary antibody (ABE1032, Millipore, Milan, Italy, 1:800 dilution) at 4 °C. After removing the primary antibodies, the slides were incubated with both Alexa Fluor 594 goat anti-rabbit (A-11012, Thermo Fisher Scientific, 1:250 dilution) and Alexa Fluor 488 goat anti-mouse (A-11001, Thermo Fisher Scientific, 1:250 dilution) secondary antibodies for 1.5 h at room temperature. After washing off the excess secondary antibodies, the slides were incubated with Hoechst, Sigma-Aldrich (1:5000 dilution) for the detection of cell nuclei. The slides were then mounted by using an anti-fade medium (Fluoromount, F4680, Sigma-Aldrich). The specificity of the immunofluorescent staining for Aβ/APP and PDIA3 was confirmed on a separate set of slides by processing the brain slices as previously described and by excluding the incubation with the primary antibodies. The slices were observed under a Nikon 80i Eclipse microscope equipped with a Qicam 12-bit Fast 1394 digital camera, and NIS-elements BR software (Nikon, Tokyo, Japan), and images were acquired for the semi-quantitative analyses of Aβ/APP and PDIA3 expression, which were performed by using the freeware software ImageJ 1.45 s, and were expressed as optical densities.

### 4.4. Triple-Immunofluorescence

For the triple fluorescence immunostaining, each 20-µm-thick brain coronal section was mounted on a slide and stored at −20 °C. After several washes with PBS, and then with 0.1% Triton X-100 in PBS (PBS*/*Triton 0.1%), the slides were incubated for 10 min into a sodium citrate buffer (10 mM sodium citrate/0.05% Triton X-100, pH 6) pre-heated at 95 °C for antigen retrieval. The slides were then cooled at room temperature in a water bath and washed with PBS/Triton 0.1%. Before the incubation with primary antibodies, the brain sections were incubated in a blocking solution containing 10% BSA and 0.3% Triton X-100 in PBS (PBS/Triton 0.3%). Thereafter, the slides were incubated for 16 h at 4 °C with the three primary antibodies: anti-GFAP chicken polyclonal antibody (ab4674, Abcam, Cambridge, UK, 1:1000 dilution), anti-NeuN mouse monoclonal antibody (ab104224, Abcam, 1:1000 dilution), and anti-PDIA3 rabbit polyclonal antibody (1:800 dilution) diluted in a solution containing 10% BSA in PBS/Triton 0.3%. After washing off the excess antibodies, the sections were incubated with secondary antibodies: DyLight 350 goat anti-chicken (SA5-10069, Thermo Fisher Scientific, 1:250 dilution), Alexa Fluor 488 goat anti-mouse (1:700 dilution), and Alexa Fluor 594 goat anti-rabbit (1:700 dilution) for 1.5 h at room temperature. After washing off the excess secondary antibodies, the slides were mounted with the anti-fade medium. The specificity of the immunofluorescent staining for GFAP, NeuN and PDIA3 was confirmed as described in the previous paragraph. The slices were then observed under a Nikon 80i Eclipse microscope equipped with a Qicam 12-bit Fast 1394 digital camera, and NIS-elements BR software (Nikon, Tokyo, Japan).

### 4.5. Statistical Analysis

The Aβ and PDIA3 optical density values were analyzed by two-way ANOVA, with genotype (3×Tg-AD vs. Non-Tg) and age (6 months of age vs. 18 months of age) as between-subject factors. Tukey’s Honestly Significant Difference (HSD) test was used for multiple post hoc comparisons, when required. The statistical significance threshold was set at *p* < 0.05.

The correlation analysis between Aβ/APP and PDIA3 protein levels was performed on the respective optical densities measured on double immunofluorescent slices and expressed as a percentage of those measured in 6-month-old 3×Tg-AD mice, by using the Pearson correlation test. These analyses were performed by using the SPSS STATISTICS software version 22.

## 5. Conclusions

The results of our study demonstrate that PDIA3 levels in the limbic regions of both 6- and 18-month-old Non-Tg and 3×Tg-AD mice are modulated in an age- and pathology-dependent fashion. Moreover, by analyzing the expression pattern of PDIA3 in 6- and 18-month-old 3×Tg-AD mice, we observed, for the first time, the expression of PDIA3 in differentiated neurons and astrocytes from the basolateral amygdala, entorhinal cortex, and dorsal and ventral CA1 regions of the hippocampus.

To our knowledge, this is the first study demonstrating that PDIA3 has a dual expression profile in 3×Tg-AD mice, probably due to its different modulation during the progression of Aβ and tau pathology developed by this AD model. In particular, during the mild phase of AD-like pathology (6 months of age) the reduced levels of PDIA3 might be associated with the decreased capacity of the ER to process the intraneuronal Aβ immunoreactivity; conversely, in the late phase of the disease (18 months of age) the increased levels of PDIA3 might be related to the progression of AD.

The age-dependent increase in ROS and NO levels can lead to protein oxidation, and specific cellular degrading systems play a role in the removal of the oxidized proteins [83]. PDIA3 modifications hamper protein redox and chaperone activities and, moreover, a proteasome-dependent turnover of protein disulfide isomerase in oxidatively stressed cells has been reported [84]. The consequent decreased capacity of the ER chaperones to process newly synthesized membrane and secretory proteins could lead to the accumulation of misfolded proteins, providing a basis for many senescence-associated alterations, including neuronal cell death. On the other hand, PDIA3 can up-regulate the mTOR signaling pathway [73]. Additionally, the increase in Aβ developed with AD pathology can lead/leads to up-regulation of the mTOR signaling pathway and could promote, as a stress response, the expression of PDIA3 and its redox and chaperone activities. An increased level of PDIA3 has been also observed in microglial and neuronal cells stimulated by Aβ [38,39], and prolonged mTORC1 activation might cause metabolic dysregulation [85]. PERK is one of the three ER stress sensors on the ER membrane. Although PERK is a controversial target in the context of neurodegenerative diseases, PDIA3 has been reported as an inhibitor of the PERK signaling pathway, suppressing PERK activation via PDI reduction [86]. As stated above, the inhibition/suppression of PDIA3, leading to conditional stimulations of the PERK signaling pathway, showed beneficial effects on mice with tauopathies [80].

Consistent with these notions, it is possible that the accumulation of misfolded proteins might fuel aging processes by modulating the mTORC1 signaling pathway [72]. In these conditions, PDIA3 will accumulate with Aβ in neuronal cells during the progression of AD, leading to chronic inflammation and neuronal cell apoptosis. This is in agreement with the observation of PDIA3-positive microglia cells, generally recruited during chronic inflammatory processes, in the limbic regions of both 6- and 18-month-old 3×Tg-AD mice. Therefore, during the mild stage of AD the use of PDIA3-positive modulators, such as diosgenin [36], which stimulates the expression and/or the activity of PDIA3, may delay the onset of the pathology. On the other hand, the use of PDIA3 inhibitors, such as punicalagin [87], in the late phase of AD, might reduce the PDIA3-dependent pro-apoptotic effects and thus slow down the disease progression. Very recently it has been reported that 16F16, a PDIA3 inhibitor, enhances the antiproliferative effect of the mTOR inhibitor everolimus in liver cancer [88].

This is a preliminary study aimed at analyzing the distribution of PDIA3 in several brain areas and correlating this with AD-like pathological hallmarks. Despite the limited observations within the time windows in various phases of pathology, and the use of a single genetic mouse model of AD, this work provides information on the multiple roles of PDIA3 and opens a new perspective on its relationship with neurological diseases, suggesting PDIA3 as a potentially valid therapeutic target.

The comprehension of the early molecular mechanisms involved in AD etiopathogenesis is fundamental to developing new prevention strategies and to improving therapeutic options. Despite the limits of our preclinical experimental study, and avoiding any simplistic extrapolation of data from the animal model to the human condition, the results of this research suggest that the pharmacological modulation of PDIA3 may have an important impact on the onset and/or progression of AD.

## Figures and Tables

**Figure 1 ijms-24-03005-f001:**
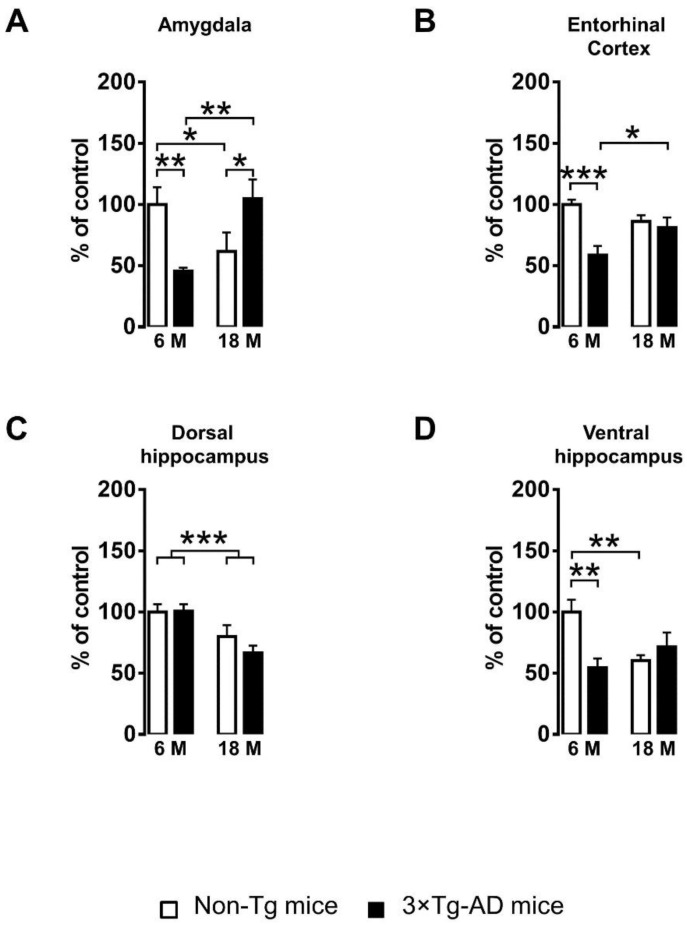
PDIA3 protein expression in Non-Tg and 3×Tg-AD mice. PDIA3 protein levels were detected by dot blot assay in different brain regions of the limbic system collected from 6- and 18-month-old Non-Tg and 3×Tg-AD mice. The limbic regions considered include: (**A**) amygdala; (**B**) entorhinal cortex; (**C**) dorsal hippocampus; (**D**) ventral hippocampus. Protein levels of all mice were normalized to those measured in 6-month-old Non-Tg mice, which were used as a control group. Data are expressed as means ± SEM. All data were analyzed by two-way ANOVA; Tukey’s test was used as a post hoc test to perform multiple comparisons (* *p* < 0.05; ** *p* < 0.01; *** *p* < 0.001).

**Figure 2 ijms-24-03005-f002:**
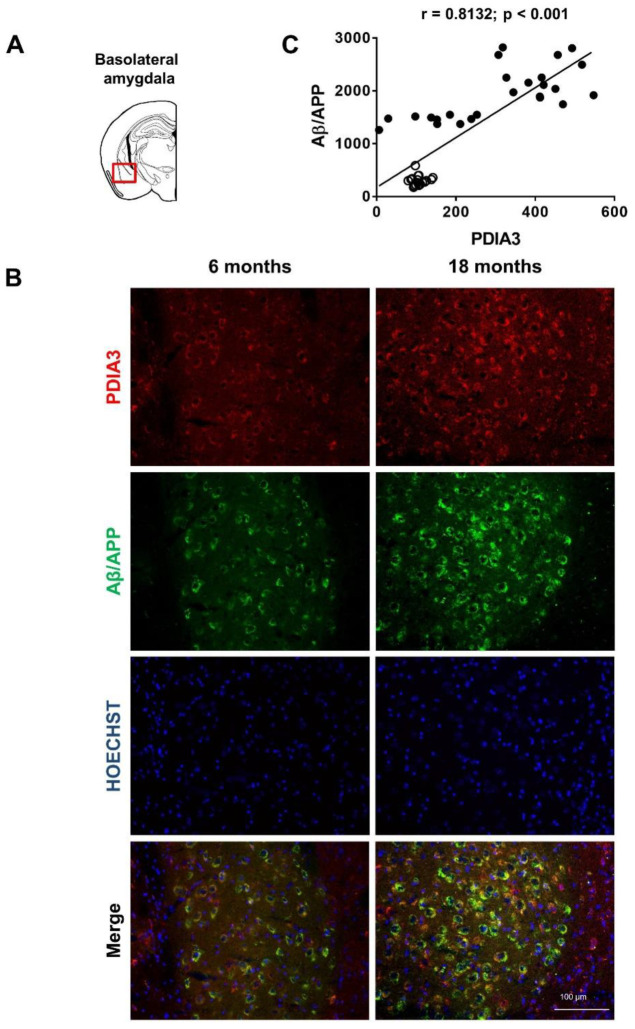
Aβ/APP-PDIA3 protein levels observed in the basolateral amygdala of 3×Tg-AD mice: (**A**) The brain diagram illustrates the site where the representative microphotographs of the basolateral amygdala were taken; (**B**) Representative microphotographs of Aβ/APP-PDIA3 double-fluorescent immunostaining (green and red, respectively) performed on brain slices of basolateral amygdala collected from 6- and 18-month-old 3×Tg-AD mice. Hoechst (blue) was used as fluorescent counterstain. Original magnification: 20×; scale bar was set at 100 µm; (**C**) Scatterplot of Aβ/APP vs. PDIA3 protein levels in the basolateral amygdala of 6- and 18-month-old 3×Tg-AD mice. Aβ/APP and PDIA3 levels refers to the results obtained by the semiquantitative analyses of the fluorescence signals as optical densities within the same cell and under the same exposure conditions. Open circles: 6-month-old 3×Tg-AD mice; closed circles: 18-month-old 3×Tg-AD mice. Data are expressed as mean optical densities ± SEM. Statistical significance threshold was set at *p* < 0.05.

**Figure 3 ijms-24-03005-f003:**
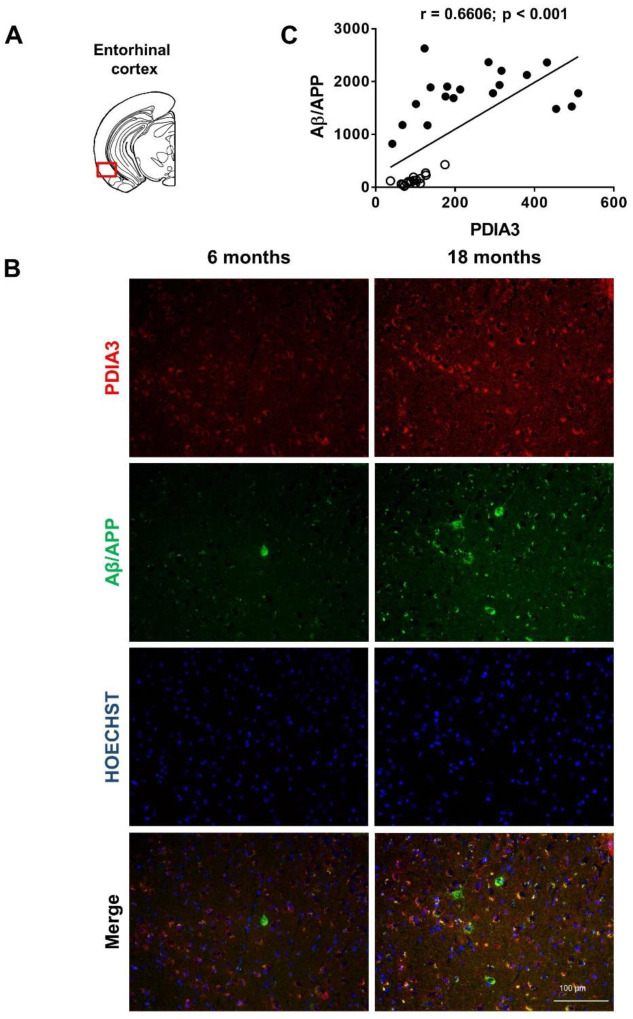
Aβ/APP-PDIA3 protein levels observed in the entorhinal cortex of 3×Tg-AD mice: (**A**) The brain diagram illustrates the site where the representative microphotographs of the entorhinal cortex were taken; (**B**) Representative microphotographs of Aβ/APP-PDIA3 double-fluorescent immunostaining (green and red, respectively) performed on brain slices of entorhinal cortex collected from 6- and 18-month-old 3×Tg-AD mice. Hoechst (blue) was used as fluorescent counterstain. Original magnification: 20×; scale bar was set at 100 µm; (**C**) Scatterplot of Aβ/APP vs. PDIA3 protein levels in the entorhinal cortex of 6- and 18-month-old 3×Tg-AD mice. Aβ/APP and PDIA3 levels refers to the results obtained by the semiquantitative analyses of the fluorescence signals as optical densities within the same cell and under the same exposure conditions. Open circles: 6-month-old 3×Tg-AD mice; closed circles: 18-month-old 3×Tg-AD mice. Data are expressed as mean optical densities ± SEM. Statistical significance threshold was set at *p* < 0.05.

**Figure 4 ijms-24-03005-f004:**
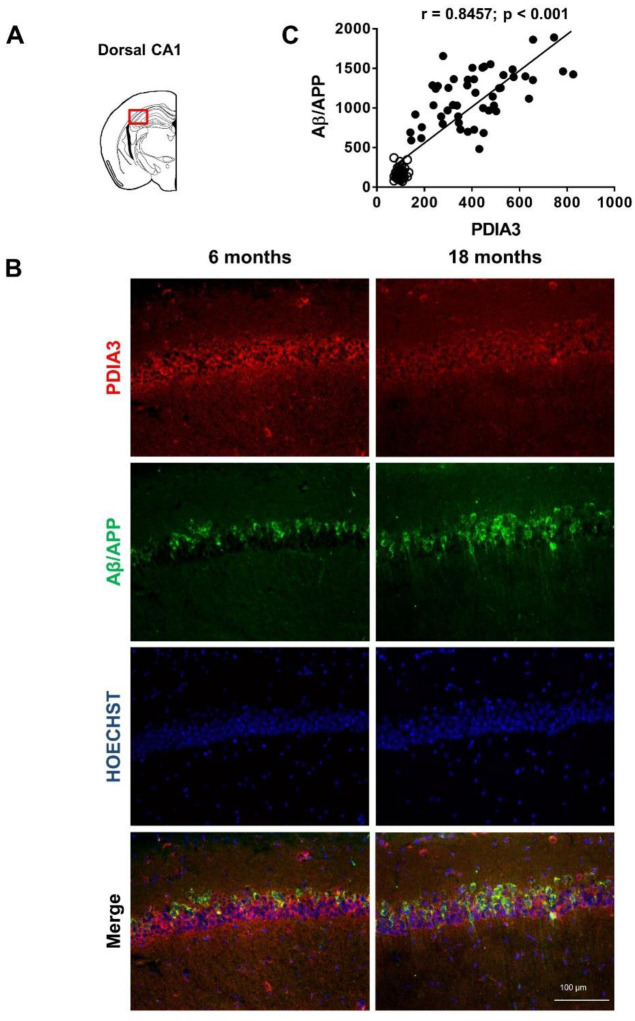
Aβ/APP-PDIA3 protein levels revealed in the dorsal CA1 region of hippocampus of 3×Tg-AD mice: (**A**) The brain diagram illustrates the site where the representative microphotographs of the dorsal CA1 region of hippocampus were taken; (**B**) Representative microphotographs of Aβ/APP-PDIA3 double-fluorescent immunostaining (green and red, respectively) performed on brain slices of dorsal CA1 region of hippocampus collected from 6- and 18-month-old 3×Tg-AD mice. Hoechst (blue) was used as fluorescent counterstain. Original magnification: 20×; scale bar was set at 100 µm; (**C**) Scatterplot of Aβ/APP vs. PDIA3 protein levels in the dorsal CA1 region of hippocampus of 6- and 18-month-old 3×Tg-AD mice. Aβ/APP and PDIA3 levels refers to the results obtained by the semiquantitative analyses of the fluorescence signals as optical densities within the same cell and under the same exposure conditions. Open circles: 6-month-old 3×Tg-AD mice; closed circles: 18-month-old 3×Tg-AD mice. Data are expressed as mean optical densities ± SEM. Statistical significance threshold was set at *p* < 0.05.

**Figure 5 ijms-24-03005-f005:**
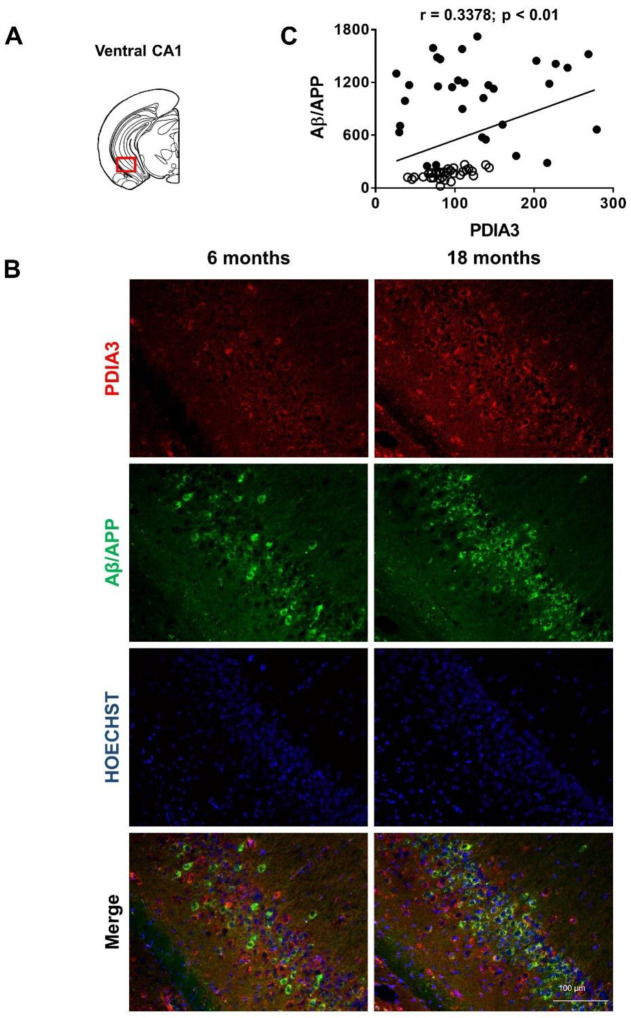
Aβ/APP-PDIA3 protein levels observed in the ventral CA1 region of hippocampus of 3×Tg-AD mice: (**A**) The brain diagram illustrates the site where the representative microphotographs of the ventral CA1 region of hippocampus were taken; (**B**) Representative microphotographs of Aβ/APP-PDIA3 double-fluorescent immunostaining (green and red, respectively) performed on brain slices of ventral CA1 region of hippocampus collected from 6- and 18-month-old 3×Tg-AD mice. Hoechst (blue) was used as fluorescent counterstain. Original magnification: 20×; scale bar was set at 100 µm; (**C**) Scatterplot of Aβ/APP vs. PDIA3 protein levels in the ventral CA1 region of hippocampus of 6- and 18-month-old 3×Tg-AD mice. Aβ/APP and PDIA3 levels refers to the results obtained by the semiquantitative analyses of the fluorescence signals as optical densities within the same cell and under the same exposure conditions. Open circles: 6-month-old 3×Tg-AD mice; closed circles: 18-month-old 3×Tg-AD mice. Data are expressed as mean optical densities ± SEM. Statistical significance threshold was set at *p* < 0.05.

**Figure 6 ijms-24-03005-f006:**
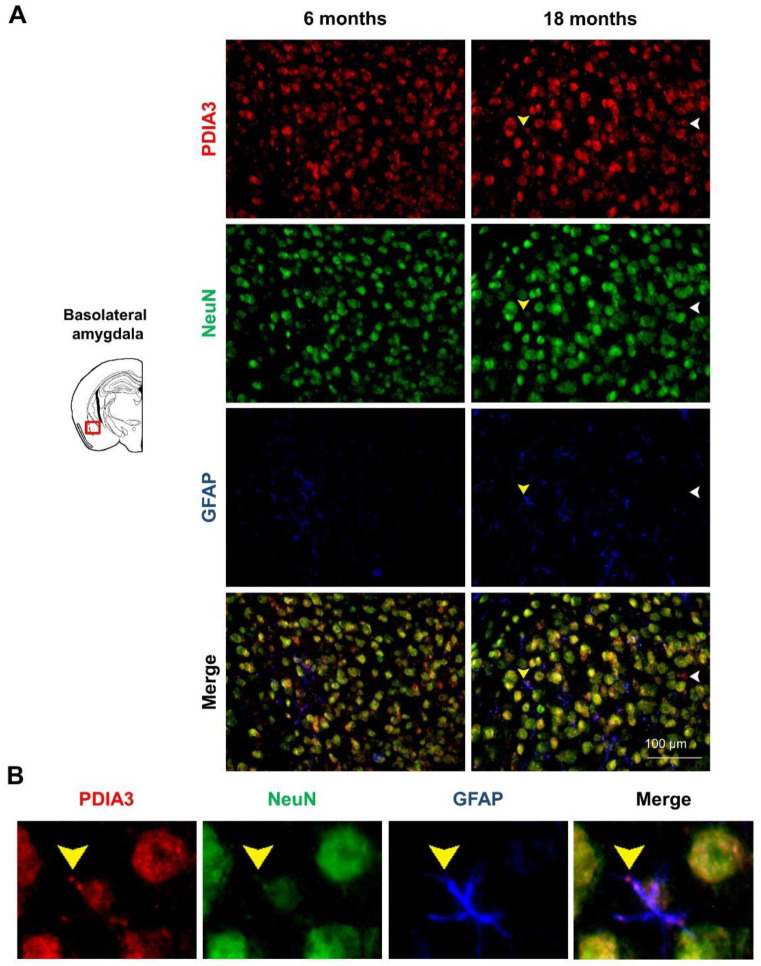
PDIA3-NeuN-GFAP protein levels observed in the basolateral amygdala of 3×Tg-AD mice: (**A**) Representative microphotographs of PDIA3-NeuN-GFAP triple-fluorescent immunostaining (red, green, and blue, respectively) performed on brain slices of the basolateral amygdala collected from 6- and 18-month-old 3×Tg-AD mice. The white arrows indicate the PDIA3 single-positive cells while the yellow arrows indicate the PDIA3-GFAP co-localization. Original magnification: 20×; scale bar was set at 100 µm; (**B**) Representative microphotographs of PDIA3-GFAP co-localization in astrocytes from basolateral amygdala of 18-month-old 3×Tg-AD mice.

**Figure 7 ijms-24-03005-f007:**
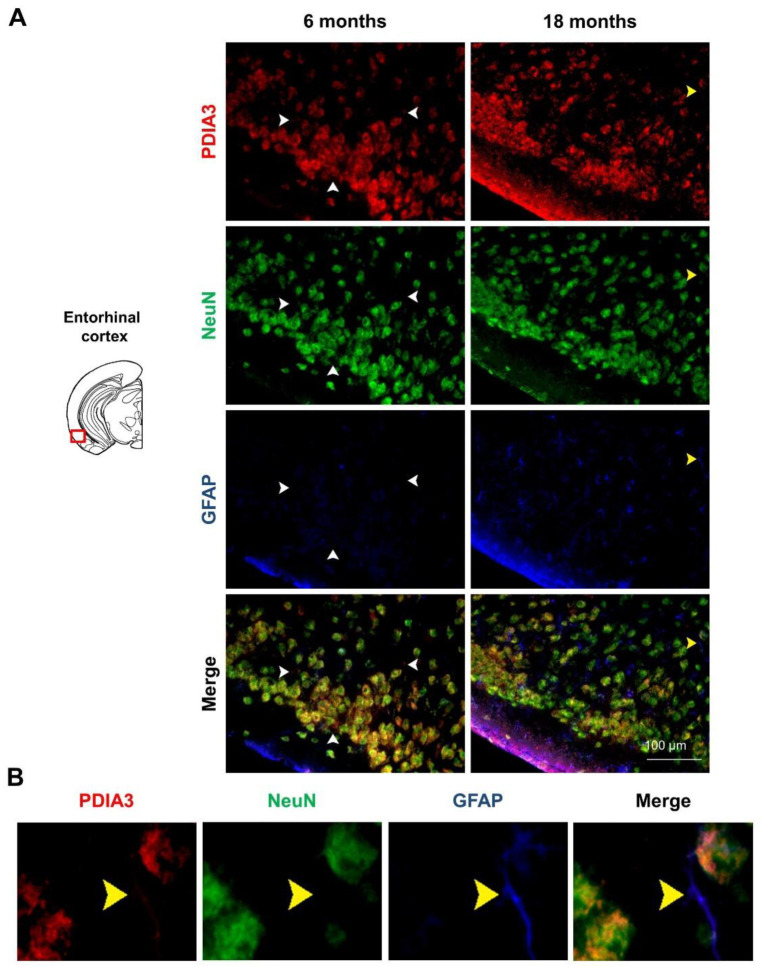
PDIA3-NeuN-GFAP protein levels observed in the entorhinal cortex of 3×Tg-AD mice: (**A**) Representative microphotographs of PDIA3-NeuN-GFAP triple-fluorescent immunostaining (red, green, and blue, respectively) performed on brain slices of entorhinal cortex collected from 6- and 18-month-old 3×Tg-AD mice. The white arrows indicate the PDIA3 single-positive cells while the yellow arrows indicate the PDIA3-GFAP co-localization. Original magnification: 20×; scale bar was set at 100 µm; (**B**) Representative microphotographs of PDIA3-GFAP co-localization in astrocytes from entorhinal cortex of 18-month-old 3×Tg-AD mice.

**Figure 8 ijms-24-03005-f008:**
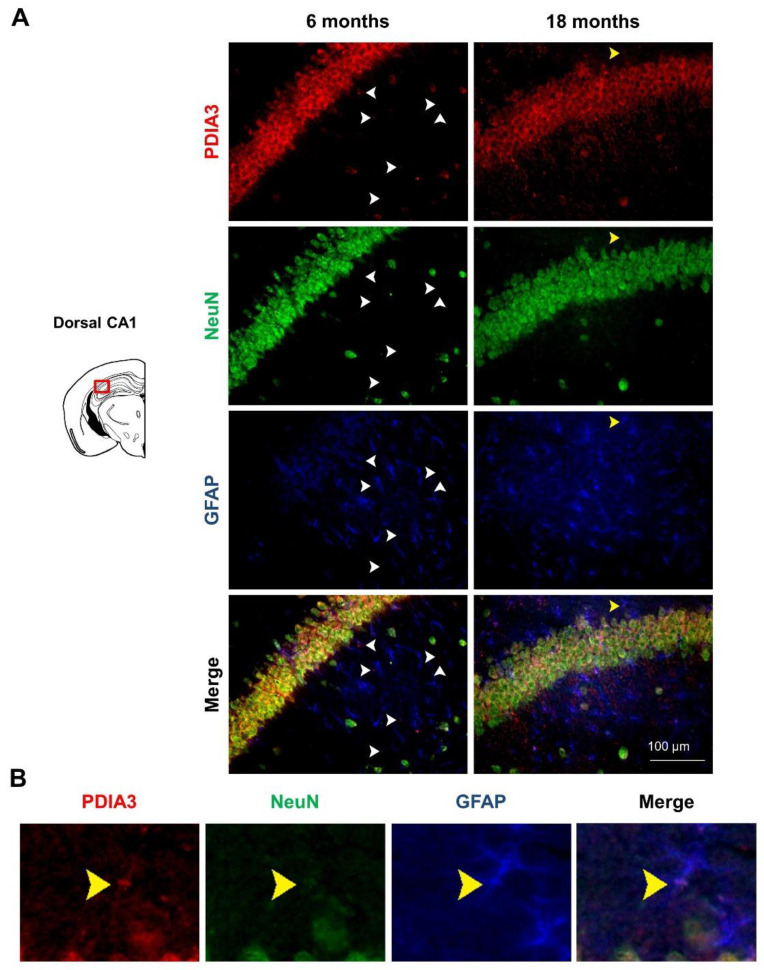
PDIA3-NeuN-GFAP protein levels observed in the dorsal CA1 region of the hippocampus of 3×Tg-AD mice: (**A**) Representative microphotographs of PDIA3-NeuN-GFAP triple-fluorescent immunostaining (red, green, and blue, respectively) performed on brain slices of dorsal CA1 region of hippocampus collected from 6- and 18-month-old 3×Tg-AD mice. The white arrows indicate the PDIA3 single-positive cells while the yellow arrows indicate the PDIA3-GFAP co-localization. Original magnification: 20×; scale bar was set at 100 µm; (**B**) Representative microphotographs of PDIA3-GFAP co-localization in astrocytes from dorsal CA1 region of hippocampus of 18-month-old 3×Tg-AD mice.

**Figure 9 ijms-24-03005-f009:**
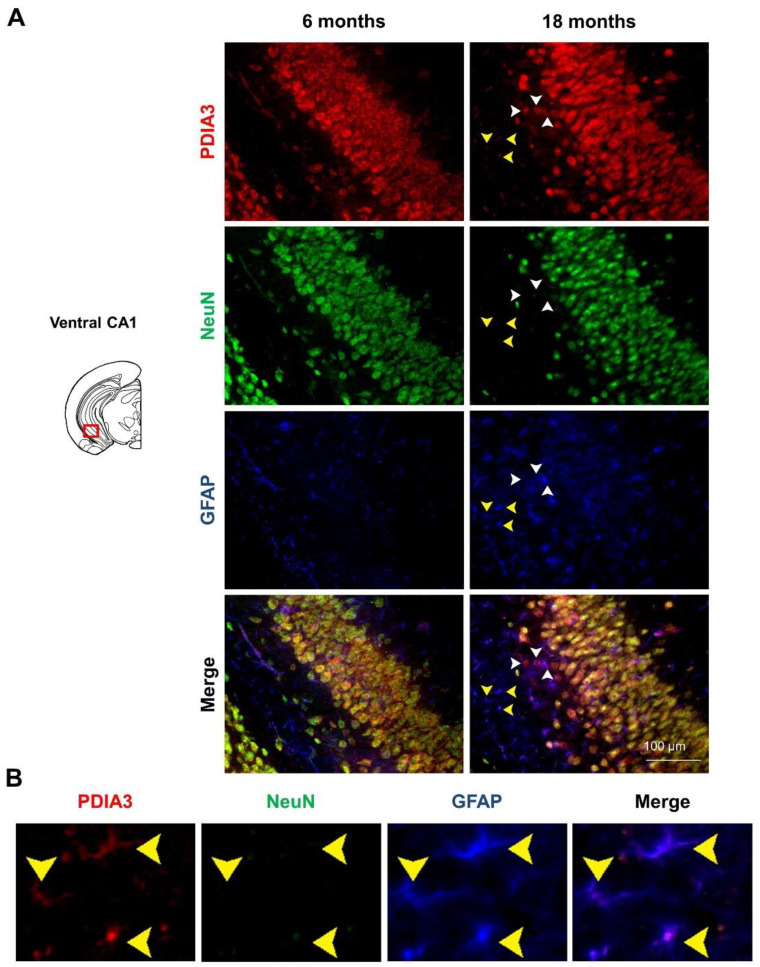
PDIA3-NeuN-GFAP protein levels observed in the ventral CA1 region of the hippocampus of 3×Tg-AD mice: (**A**) Representative microphotographs of PDIA3-NeuN-GFAP triple-fluorescent immunostaining (red, green, and blue, respectively) performed on brain slices of ventral CA1 region of hippocampus collected from 6- and 18-month-old 3×Tg-AD mice. The white arrows indicate the PDIA3 single-positive cells while the yellow arrows indicate the PDIA3-GFAP co-localization. Original magnification: 20×; scale bar was set at 100 µm; (**B**) Representative microphotographs of PDIA3-GFAP co-localization in astrocytes from ventral CA1 region of hippocampus of 18-month-old 3×Tg-AD mice.

**Table 1 ijms-24-03005-t001:** Results from the statistical analysis of data obtained from the dot blot analysis.

Brain Region	Genotype (G)	Age (A)	Interaction (G × A)
Amygdala	F_(1,35)_ = 0.192, n.s.	F_(1,35)_ = 0.622, n.s.	F_(1,35)_ = 13.657, *p* < 0.001
Entorhinal cortex	F_(1,35)_ = 12.781, *p* < 0.01	F_(1,35)_ = 0.433, n.s.	F_(1,35)_ = 7.787, *p* < 0.01
Dorsal hippocampus	F_(1,35)_ = 0.826, n.s.	F_(1,35)_ = 14.961, *p* < 0.001	F_(1,35)_ = 1.013, n.s.
Ventral hippocampus	F_(1,35)_ = 3.789, n.s.	F_(1,35)_ = 1.619, n.s.	F_(1,35)_ = 10.287, *p* < 0.01

Two-way analysis of variance (ANOVA) with genotype (3×Tg-AD vs. Non-Tg) and age (6 vs. 18 months of age) as between-subject factors (*n* = 3 per group). F_(1,35)_ = F value with (1,35) degrees of freedom; n.s. = not significant.

**Table 2 ijms-24-03005-t002:** Results from the Pearson’s correlation analysis of data obtained from the double immunofluorescence analysis.

	Amygdala	Entorhinal Cortex	Dorsal Hippocampus	Ventral Hippocampus
Pearson correlation coefficient (r)	0.8132	0.6606	0.8457	0.3378
*p* value	<0.001	<0.001	<0.001	<0.01

Pearson’s correlation analysis (PDIA3 vs. Aβ) in 3×Tg-AD mice of 6- and 18-month-old (*n* = 3 per group). (r): Pearson correlation coefficient.

## Data Availability

Data are contained within the article.

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
