# Peer review of "PDIA3 Expression Is Altered in the Limbic Brain Regions of Triple-Transgenic Mouse Model of Alzheimer’s Disease"

_ijms, 2023, doi:10.3390/ijms24033005_

Round 1

Reviewer 1 Report

In this manuscript, the authors assess the levels of PDIA3 as a function of age across three brain regions in a mouse model of AD.

The work is interesting and contributes to the overall knowledge of the field. The experimental design is straightforward and very well executed. The data presented are of high quality and support the authors’ conclusions.

I would suggest the following:

·         The authors should expand the discussion about the implications of the data showing that in the amygdala, entorhinal cortex, and dorsal hippocampus the levels of PDIA3 are already reduced in 3xTg-AD mice at 6 months of age.

 ·         It is surprising that the authors did not address a possible correlation between tau accumulation/phosphorylation and PDIA3 levels. This would increase the impact of the manuscript.

Author Response

Reviewer 1

The work is interesting and contributes to the overall knowledge of the field. The experimental design is straightforward and very well executed. The data presented are of high quality and support the authors’ conclusions.

Dear Reviewer,

Thank you for reviewing our manuscript. We appreciate your helpful and valuable suggestions. We have reviewed them carefully and believe we have made all appropriate changes to our manuscript.

I would suggest the following:

  • The authors should expand the discussion about the implications of the data showing that in the amygdala, entorhinal cortex, and dorsal hippocampus the levels of PDIA3 are already reduced in 3xTg-AD mice at 6 months of age.

The following sentence has been added to discussion:

“A possible explanation for the observed reduced level of PDIA3 in the limbic brain regions of 3×Tg-AD mice at 6 months of age, as compared to age-matched Non-Tg mice, is the role of PDIA3 as a molecular carrier for Aβ monomers, preventing the formation of in-soluble aggregates. Aβ peptides produced during the APP processing and sequestered by PDIA3 and calreticulin chaperones can be eliminated through the cerebrospinal fluid [11] thus reducing PDIA3 level. Furthermore, it has been showed that short-term Aβ25-35 treatment of human neuroblastoma cells induces PDIA3 decrease in intracellular protein levels, different intracellular localization, and PDIA3 secretion in the cultured medium [39].”

  • It is surprising that the authors did not address a possible correlation between tau accumulation/phosphorylation and PDIA3 levels. This would increase the impact of the manuscript.

The following sentences have been added to discussion and conclusions sections:

“An association between several ER stress markers as well as unfolded protein response (UPR) proteins and the accumulation of NFTs has been observed in post‐mortem brain tissues from tauopathy patients, with a positive relationship between the severity of protein aggregation and disease status [79]. Recently, it has been reported that a small molecule, SB1617, can suppress abnormal tau protein aggregation through PDIA3 inhibition and the enhancement of protein kinase-like endoplasmic reticulum kinase (PERK) signaling pathway [80]. NFTs accumulation occurs during late stage of AD pathology. Thus, the inhibition of PDIA3 may be an effective strategy for regulating tauopathies and modulate AD progression.”

“PERK is one of the three ER stress sensors on the ER membrane. Although PERK is a controversial target in the context of neurodegenerative diseases, PDIA3 has been reported as inhibitor of the PERK signaling pathway suppressing PERK activation via PDI reduction [88]. As above stated, the inhibition/suppression of PDIA3, leading to conditional stimulations of PERK signaling pathway, showed beneficial effects on mice with tauopathies [80].”

Reviewer 2 Report

In the manuscript entitled ‘PDIA3 expression is altered in the limbic brain regions of triple-transgenic mouse model of Alzheimer's disease’, Cassano et al. report age- and/or pathology-dependent changes in PDIA3 levels in 3×Tg-AD mice, suggesting that PDIA3 is a potential therapeutic target for the treatment of AD. Although this study is generally well-prepared, some improvements can still be made with the following suggestions/comments.

1.       The main data are based on immunostaining investigations, which are prone to subjective quantitation. Henceforth, I strongly recommend the authors to further support their conclusions with other methods.

2.       The authors merely investigated the expression of PDIA3 in 3×Tg-AD mice at 6 and 18 months of age. It is surprising to see that the expression profiles are not consistent between time windows. I, or probably most of the readers, would like to know PDIA3 expression changes in various phases of pathogenesis in multiple AD models.

3.       Please provide high magnification confocal images with orthogonal views for Figures 6B, 7B, 8B and 9B.

4.       The authors should quantify images in Figures 6-9.

5.       The authors state that PDIA3 is expressed in some microglial cells, but supporting data are sadly lacking.

6.       The authors hypothesized that the significant reduction of PDIA3 expression in the limbic brain regions of 3×Tg-AD mice at 6 months of age may underlie and/or contribute to the formation of intracellular Aβ oligomers. Unfortunately, the authors did not test such an interesting hypothesis.

7.       It would be great to discuss the limitation of the present study as well as the ways to improve translation from preclinical findings to clinical trials in AD patients.

Author Response

Reviewer 2

In the manuscript entitled ‘PDIA3 expression is altered in the limbic brain regions of triple-transgenic mouse model of Alzheimer's disease’, Cassano et al. report age- and/or pathology-dependent changes in PDIA3 levels in 3×Tg-AD mice, suggesting that PDIA3 is a potential therapeutic target for the treatment of AD. Although this study is generally well-prepared, some improvements can still be made with the following suggestions/comments.

Dear Reviewer,

Thank you for reviewing our manuscript. We appreciate your helpful and valuable suggestions. We have reviewed them carefully and believe we have made all appropriate changes to our manuscript.

  1. The main data are based on immunostaining investigations, which are prone to subjective quantitation. Henceforth, I strongly recommend the authors to further support their conclusions with other methods.

We agree with reviewer’s worries about the immunostaining investigation. However, we have tried to carry out the analyses with the utmost care by processing the samples from each experimental group together and by evaluating the signals of all the samples under the same microscopy conditions in terms of exposure and gain. Additionally, immunofluorescence analyses have been carried out evaluating the Abeta/APP and PDIA3 signals within the same cells, under the same exposure and gain, and this could be impossible by a blotting analysis on protein extracts. Pearson correlation test was performed only on 3xTg-AD mice because they represent the best model suitable to evaluate the relationship of PDIA3 pattern of expression with AD progression.

  1. The authors merely investigated the expression of PDIA3 in 3×Tg-AD mice at 6 and 18 months of age. It is surprising to see that the expression profiles are not consistent between time windows. I, or probably most of the readers, would like to know PDIA3 expression changes in various phases of pathogenesis in multiple AD models.

We agree with referee that similar investigations might be useful in other experimental models of AD and probably the eventual publication of the present study will stimulate such possibility, however, testing in other models was beyond the aim of our study. Moreover, we decided to focus on two different stages of the 3×Tg-AD phenotype including young 3×Tg-AD mice, which do not show overt sign of AD-like pathological conditions and old mice, which show all the AD-like neurofunctional and behavioral alterations allowed us to reduce the number of animals involved in the study. With these two extreme phases of AD-like development we opened a new perspective that PDIA3 might play a role in AD, as suggested by the strong correlations observed. Using other animal models or other ages of our model would significantly increase the number of animals involved, which, at the stage of testing a novel hypothesis, we evaluated as being unethical.

  1. Please provide high magnification confocal images with orthogonal views for Figures 6B, 7B, 8B and 9B.

Slices were then observed under a Nikon 80i Eclipse microscope equipped with a Qicam 12-bit Fast 1394 digital camera, and NIS-elements BR software (Nikon, Tokyo, Japan). No confocal images were collected.

  1. The authors should quantify images in Figures 6-9.

The triple fluorescence immunostaining was performed to investigate the expression of PDIA3 within different cells of the selected brain areas analyzed. To this purpose we used anti-NeuN for neuronal cells and anti-GFAP for astrocyte cells. In this way we observed that PDIA3 resulted expressed in differentiated neurons in all brain regions of 3×Tg-AD mice at both time points considered

  1. The authors state that PDIA3 is expressed in some microglial cells, but supporting data are sadly lacking.

We based our statement on cell staining with specific markers and observing cellular morphology. We hypothesized that those cells could be microglial cells and/or immature/suffering neurons (Lavezzi et al 2014, ref 48). Anyway, in keeping with reviewer’s comment, we modified the text as follows: “Interestingly, we also observed PDIA3-positive staining in a number of GFAP- and NeuN-negative cells from all the limbic areas considered in this study (Figures 5-9 A; white arrows). Based on the morphology and dimension of PDIA3-positive cells, we hypothesized that these cells may refer to microglial cells and/or immature/suffering neurons. To this regard, it has been previously demonstrated that NeuN staining can be altered or lost in immature and/or suffering neurons [48]. Future studies are needed to confirm such hypothesis”. Moreover, we removed such information, which at the moment is poorly supported by our experimental data, from the abstract.

  1. The authors hypothesized that the significant reduction of PDIA3 expression in the limbic brain regions of 3×Tg-AD mice at 6 months of age may underlie and/or contribute to the formation of intracellular Aβ Unfortunately, the authors did not test such an interesting hypothesis.

The following sentence has been added to discussion:

“A possible explanation for the observed reduced level of PDIA3 in the limbic brain regions of 3×Tg-AD mice at 6 months of age, as compared to age-matched Non-Tg mice, is the role of PDIA3 as a molecular carrier for Aβ monomers, preventing the formation of in-soluble aggregates. Aβ peptides produced during the APP processing and sequestered by PDIA3 and calreticulin chaperones can be eliminated through the cerebrospinal fluid [11] thus reducing PDIA3 level. Furthermore, it has been showed that short-term Aβ25-35 treatment of human neuroblastoma cells, induces PDIA3 decrease in intracellular protein levels, different intracellular localization, and PDIA3 secretion in the cultured medium [39].”

  1. It would be great to discuss the limitation of the present study as well as the ways to improve translation from preclinical findings to clinical trials in AD patients.

The following sentence has been added to end of conclusions:

“This is a preliminary study aimed to analyze the distribution of PDIA3 in several brain area and correlate this with AD-like pathological hallmarks. Although the limited observation within the time windows in various phases of pathology and the use of a single genetic mouse model of AD, this work provides information on the multiple roles of PDIA3 and open a new perspective on its relationship with neurological diseases suggesting PDIA3 as a potentially valid therapeutical target.”

Reviewer 3 Report

This study demonstrates that PDIA3 levels in the limbic regions of both 6- and 18-month-old Non-Tg and 3×Tg-AD mice are modulated in an age- and pathology- dependent fashion. At the same time, the expression of PDIA3 in the differentiated neurons and astrocytes of the basolateral amygdala, the inner olfactory cortex, the dorsal and ventral CA1 regions was observed. The article is well organized and its presentation is good, which deserve publication in International Journal. of Molecular. Sciences. However, some minor issues still need to be improved:

1.     Table 1 and Table 1 are not described in detail in the text, the table 1 notes do not indicate what F (1,35) and n.s. refer to.

2.     “…a direct correlation between the level of Aβ and PDIA3 proteins in all the brain regions analysed, and the presence of PDIA3 in all post-mitotic neurons and astrocytes, as well as in some microglial cells and/or immature/suffering neurons.” This sentence in the abstract has not been proven in the text. Please provide the basis.

Author Response

Reviewer 3

This study demonstrates that PDIA3 levels in the limbic regions of both 6- and 18-month-old Non-Tg and 3×Tg-AD mice are modulated in an age- and pathology- dependent fashion. At the same time, the expression of PDIA3 in the differentiated neurons and astrocytes of the basolateral amygdala, the inner olfactory cortex, the dorsal and ventral CA1 regions was observed. The article is well organized and its presentation is good, which deserve publication in International Journal. of Molecular. Sciences. However, some minor issues still need to be improved:

Dear Reviewer,

Thank you for reviewing our manuscript. We appreciate your helpful and valuable suggestions. We have reviewed them carefully and believe we have made all appropriate changes to our manuscript.

  1. Table 1 and Table 1 are not described in detail in the text, the table 1 notes do not indicate what F (1,35)and n.s. refer to.

F (1,35) represents the F value with (1,35) degrees of freedom; n.s. stands for not significant. (r) is the Pearson correlation coefficient. This information has been added to Table 1 and 2 footnotes.

  1. “…a direct correlation between the level of Aβ and PDIA3 proteins in all the brain regions analyzed, and the presence of PDIA3 in all post-mitotic neurons and astrocytes, as well as in some microglial cells and/or immature/suffering neurons.” This sentence in the abstract has not been proven in the text. Please provide the basis.

We agree with reviewer’s comment and modified the abstract, as follows:

“A comparative immunohistochemistry analysis performed on 3×Tg-AD mice at 6 (mild AD-like pathology) and 18 (severe AD-like pathology) months of age showed a direct correlation between the cellular level of Aβ and PDIA3 proteins in all the brain regions analyzed, even if with different magnitudes.  Additionally, immunohistochemistry analysis showed the presence of PDIA3 in all post-mitotic neurons and astrocytes. Overall, altered PDIA3 levels appear to be age- and/or pathology-dependent, corroborating the ER chaperones involvement in AD pathology, and supporting the PDIA3 protein as a potential novel therapeutic target for the treatment of AD.”

Round 2

Reviewer 1 Report

The authors have successfully addressed my previous comments.

Reviewer 2 Report

I have no more concerns.